# Use of a Silicon Microneedle Chip-Based Device for the Extraction and Subsequent Analysis of Dermal Interstitial Fluid in Heart Failure Patients

**DOI:** 10.3390/diagnostics15080989

**Published:** 2025-04-13

**Authors:** Markus Renlund, Laurenz Kopp Fernandes, Pelle Rangsten, Mikael Hillmering, Sara Mosel, Ziad Issa, Volkmar Falk, Alexander Meyer, Felix Schoenrath

**Affiliations:** 1Ascilion AB, 164 40 Kista, Sweden; 2Department of Cardiothoracic and Vascular Surgery, Deutsches Herzzentrum der Charité (DHZC), Augustenburger Platz 1, 13353 Berlin, Germanyfelix.schoenrath@dhzc-charite.de (F.S.); 3Charité—Universitaetsmedizin Berlin, Corporate Member of Freie Universitaet Berlin, Humboldt-Universitaet zu Berlin, and Berlin Institute of Health, Augustenburger Platz 1, 13353 Berlin, Germany; 4German Center for Cardiovascular Research (DZHK), Partnersite Berlin, 10785 Berlin, Germany; 5Department Health Science and Technology, Eidgenoessische Technische Hochschule (ETH) Zurich, 8092 Zurich, Switzerland

**Keywords:** microneedle, dermal interstitial fluid, biomarkers, high-sensitivity CRP, NT-proBNP, lactate, biomarker kinetics, inflammation, heart failure, micro-electro-mechanical systems (MEMS)

## Abstract

**Background/Objectives:** Dermal interstitial fluid (dISF) is probably the most interesting biofluid for biomarker analysis as an alternative to blood, enabling higher patient comfort and closer or even continuous biomarker monitoring. The prerequisite for dISF-based analysis tools is having convenient access to dISF, as well as a better knowledge of the presence, concentration, and dynamics of biomarkers in dISF. Hollow microneedles represent one of the most promising platforms for access to pure dISF, enabling the mining of biomarker information. **Methods and Results:** Here, a microneedle-based method for dISF sampling is presented, where a combination of hollow microneedles and sub-pressure is used to optimize both penetration depth in skin and dermal interstitial fluid sampling volumes, and the design of an open, prospective, exploratory, and interventional study to examine the detectability of inflammatory and cardiocirculatory biomarkers in the dISF of heart failure patients, the relationship between dISF-derived and blood-derived biomarker levels, and their kinetics during a cardiopulmonary exercise test (CPET) is introduced. **Conclusions:** The dISF sampling method and study presented here will foster research on biomarkers in dISF in general and in heart failure patients in particular. The study is part of the European project DIGIPREDICT—Digital Edge AI-deployed DIGItal Twins for PREDICTing disease progression and the need for early intervention in infectious and cardiovascular diseases beyond COVID-19.

## 1. Introduction

Biomarkers are a cornerstone of diagnosis and treatment monitoring in the vast majority of diseases. Blood is the standard biomaterial for biomarker analysis due to established sampling techniques and having plenty of experience with well validated values. Given the invasiveness of blood sampling, other biofluids like sweat, tears, and dermal interstitial fluid (dISF) move in the center of interest for biomarker analysis, enabling higher patient comfort, especially in vulnerable groups like children, and allowing for closer or even continuous monitoring, which is a crucial step towards personalized medicine. dISF has close contact to blood plasma via dermal capillaries. Blood biomarkers reach dISF paracellularly, transcellularly, and via transcytosis, resulting in some cases of there being similar biomarker levels in both dISF and blood [1]. Therefore, dISF is probably the most interesting alternative biofluid for biomarker analysis. It has previously been shown that some biomolecules have similar and some have even higher prevalence in dermal interstitial fluid compared to blood plasma [2,3], but overall, there is only limited data on dISF biomarkers so far.

Historically, it has been difficult to access dISF. Researchers have been limited to the investigation of biofluids from microdialysis [4], fluid from suction blisters [5], or reverse iontophoresis [6]. Other options to access dISF include wicking [7], squeezing, or centrifuging the fluid out of biopsies [8], or fluid collection after tape stripping [9]. However, all of these methods have limitations like invasiveness and limited patient and user comfort, and none are appropriate to provide pure dISF, but only dISF proxies.

Thus, to be able to use dISF-derived biomarkers in clinical practice, there is a great need for new extraction methods able to deliver pure dISF, as well as for studies that increase the knowledge about these biomarkers. Here, we present a novel medical device containing microneedles for the sampling of dermal interstitial fluid, and the protocol of an exploratory proof-of-concept clinical study aiming to detect cardiovascular biomarkers in the dISF of heart failure patients.

## 2. Materials and Methods

### 2.1. Sampling of Dermal Interstitial Fluid Using Microneedles

Microneedles have been available, primarily for drug delivery [10], for several decades, and during the last decade the interest in using them for extraction has started to grow [11]. One important benefit with microneedles is that they are perceived as less invasive, painful, and stigmatized by the patient compared to blood sampling and other means of accessing dISF [12]. Microneedles may be designed in many different shapes and materials depending on their intended use [13]. The major morphology types of microneedles used for extraction of dISF are solid, porous (for absorption, swelling, or hydrogel), and hollow microneedles. They can be manufactured in metals, silicon, glass, ceramics, or polymers [2,3,14,15,16,17]. Existing methods for sampling dISF using microneedles typically result in low volumes and long sampling times, thus limiting both their usability and the possibility for analysis, especially if intended for in vivo use on humans. Here, a hollow silicon microneedle design is presented and refined for several years. In addition, a method adapted to meet the requirements set by the presented study on heart failure patents will be described. The sampling procedure consists of two phases: one skin penetration phase where microneedles create micro-pores in the skin, and a subsequent extraction phase where the dISF is extracted from the skin.

#### 2.1.1. Silicon Microneedle Chip Technology

To access the interstitial fluid-rich *dermis* layer of the skin, a fluidic pass-through channel through the outer layers (*stratum corneum* and the underlying layers of the *epidermis*) needs to be created. This simple statement can be refined into several requirements for the microneedles; they should be long enough to reach the dermis but not reach the pain receptors, be sharp and mechanically robust enough to break the skin barrier, and should preferably be used in a controlled and repeatable fashion. Since they break the skin barrier, they are regarded as being regulatory invasive and need to be sterile and made from a biocompatible material. To become commercially viable, manufacturing needs to be scalable to high-volume production with well-controlled uniform dimensions and a high yield. All the above are met by choosing monocrystalline silicon as the material [18] and micro-electro-mechanical-systems (MEMS) technology as the manufacturing method [19]. By combining silicon and MEMS, arrays of (in theory) identical microneedles can be manufactured in numbers of hundreds per square centimeter (cm^2^). The presented design has been refined for many years by investigating design parameters such as the cross section shape, dimensions, number of needles per chip, needle height, needle pitch, die size, and others. In addition to the microneedle design, complex patterns for fluidic handling can easily be integrated into the chip, and thanks to the crystallinity, the needles can be shaped with ultra-sharp edges and sloped walls to cut the skin like miniature scalpels and create a clog-free structure [19,20,21,22,23]. The selected hollow silicon microneedle design is shown in Figure 1.

#### 2.1.2. Interfaces and Usability

An important design consideration with MEMS chips are the interfaces to the macroscopic world, especially the mechanical interface. Here, the microneedle chip is mounted in a plastic holder using a biocompatible foam that also acts as a sealant to the vacuum chamber (explained later) that is formed by the microneedle chip (1. Needle Unit) and other plastic parts (2. Chamber Unit). The Chamber Unit is assembled with a sterilization filter to prevent the contamination of re-usable parts and has a transparent removable tape lid to enable the optical inspection of the skin (3. Camera) during sampling (Figure 2). 

Another important interface is how the operator interacts with the device. There should be user-friendly start and stop functions. For safety reasons, there should be interrupt or abort functionalities implemented. A graphical user interface (GUI) is preferred for more advanced machines and equipment where several parameters need to be monitored for the verification of functionality or trouble shooting.

#### 2.1.3. Sub-Pressure-Assisted Skin Penetration

Another challenge is to control the penetration depth for optimal dISF access and to simultaneously avoid blood contamination by, e.g., puncturing the capillary loops in the dermal papillae close the epidermis. Hollow microneedles enable an interesting alternative and more precise way of penetrating the skin compared to punching the skin using force or impact speed with a solid needle. Instead of just using mechanical force and an impact speed to break the skin barrier, a sub-pressure (vacuum) can be applied to the backside of the chip and through the bore holes that lifts the skin towards the needles, instead of pushing the needles down into the skin. In this way, the penetration depth can be adjusted with the sub-pressure, which results in more uniform and well-controlled penetration. This method is further improved if the skin is slightly stretched prior to penetration, which is facilitated both by a frame on the silicon chip and on the protrusion in the supporting plastic holder (Figure 3).

#### 2.1.4. Sub-Pressure-Assisted dISF Extraction

A major difference between blood sampling and the sampling of dISF is that dISF has no relevant positive pressure and thus does not leave the body voluntarily even after skin puncture. The amount of dISF that naturally leaves the skin through pores created by microneedles is very limited. One way of increasing this volume is to stretch the skin or increase the pressure gradient to open the pores and force the interstitial fluid out. By using sub-pressure, both can be achieved. See the illustration in Figure 4.

#### 2.1.5. Design of the dISF Extraction Device (PELSA System)

The schematic of the dISF sampling system is shown in Figure 5. The microneedle chip is mounted in a customized tool that uses negative pressure for a controlled penetration depth and improved sampling rates. The tool consists of three single-use parts that are in contact with the patient, and a re-usable control system for optical inspection and pressure control. The single-use parts are a Needle Unit, a Chamber Unit, and a Skin Fixture. The Needle Unit consists of a microneedle chip held in place with an injection molded plastic part by foam tape. The foam tape acts as a gasket when the Needle Unit connects to the Chamber Unit in such a way that a vacuum chamber is formed. Via a connection kit, the vacuum chamber is coupled to the control unit, which in turn contains both hardware and software that creates, controls, and monitors a vacuum that enhances the skin penetration and the sampling of the fluid. The control unit also connects to an optical system fitted in a plastic handle that connects upstream to the Chamber Unit, enabling the real-time monitoring of the sampling area through an optical inspection window in the Chamber Unit.

#### 2.1.6. System Configuration and Phases During Sampling of dISF

To apply the described sub-pressure-assisted methods above, the configuration and usage of the system needs to be adapted and changed during different phases of the sampling of the dISF. There are three main phases: (1) penetration, (2) extraction, and (3) collection. During the penetration phase, the sub-pressure pulls the skin towards the microneedle with the Needle Unit attached. Prior to extraction, the Needle Unit is removed to allow for a larger skin stretch and for optical inspection of the skin using the camera. An optional buckle and armband (Skin Fixture) can hold the Chamber Unit in place during the extraction phase. After extraction, the dISF on the skin can be collected using a pipette and dispensed into a suitable container prior to further analysis. See Figure 6 below.

### 2.2. Examination of dISF-Derived Biomarkers in Heart Failure Patients and Comparison with Biomarker Levels in Blood

#### 2.2.1. Study Design

An open, prospective, exploratory, and interventional study is designed to examine the detectability of inflammatory (high-sensitivity C-reactive protein—hsCRP) and cardiocirculatory (N-terminal pro-brain-type natriuretic peptide—NT-proBNP, lactate, Olink^®^ Target 96 Cardiovascular II panel) biomarkers in the dISF of heart failure patients, the relationship between dISF-derived and blood-derived biomarker levels, and their kinetics during a cardiopulmonary exercise test (CPET).

The primary endpoint is the level above the limit of quantification of the investigated biomarkers in dermal interstitial fluid. There are several secondary endpoints, namely (a) dermal interstitial fluid levels and blood levels of the investigated biomarkers, (b) an increase or decrease in the levels of the investigated biomarkers in dermal interstitial fluid and in blood during exercise, (c) dermal interstitial fluid level change (to the first or previous measurement) and blood level change (to the first or previous measurement) of the investigated biomarkers during exercise, (d) evidence that the PELSA system can extract interstitial fluid from the skin, and (e) documentation of adverse events and serious adverse events.

The study protocol is approved by the responsible ethics committee and is registered at ClinicalTrials.gov (ID NCT06200636). Written informed consent has to be obtained from every study participant before inclusion. The study includes 20 adult patients suffering from heart failure, as defined in the 2021 ESC Heart Failure Guidelines [24], visiting our hospital’s heart failure and heart transplant outpatient unit.

The exclusion criteria include, among others, having tattoos, piercings, and skin disease on or near the sampling area, which could impede dISF sampling and quality, as well as patients on immunosuppressive drugs and patients on renal replacement therapy, for these conditions can have potential effects on the levels of the investigated biomarkers. For the whole list of inclusion and exclusion criteria, see Table 1.

In each participant, biomaterial sampling is performed thrice: (1) after a ≥15 min resting period (sitting or supine) before a CPET (T0), (2) instantaneously after a CPET (defined as the time between the end of exercise and the start of dISF extraction ≤ 12 min) (T1), and (3) ≥30 min after the end of exercise (T2); see Figure 7. The ≤12 min range between the end of exercise and the start of dISF sampling at T1 has been defined to find a balance between the shortest possible delay and practicability, i.e., to allow the patient to step off the ergometer and for the study personnel to prepare the skin and start the dISF extraction process. All timestamps will be documented in a worksheet to assure the quality and to enable the comparability of the results. A blood count is performed at every time from T0 to T2 to allow for the normalization of plasma volume using the Dill and Costill equation [25]:ΔPV [%] = 100 × ((Hb_pre_/Hb_post_) × (100 − Hct_post_)/100 − Hct_pre_) − 1),
where Hb_pre_ and Hb_post_ are the hemoglobin levels before and after exercise, respectively, and Hct_pre_ and Hct_post_ are the hematocrit values in %.

dISF sampling is performed at every time from T0 to T2, preferably quasi-simultaneously at two sites, usually on the proximal forearm, with a short time shift needed to start extraction on the second sampling site to increase the sampling volume per sampling time. Measurement of dISF volume is performed by the weighing of the samples using a high-precision balance (Mettler Toledo AG204, Mettler Toledo, Columbus, Ohio, USA), where the density of dISF can be assumed to be 1000 kg/m^3^ [26,27,28]. Aliquoting is performed using a 1–10 μL micropipette (Gilson Pipetman L P10LA, Gilson, Middleton, WI, USA). Blood sampling is also performed at each time from T0 to T2. Blood sampling is performed after the start of dISF sampling on the first sampling site and before the start on the second sampling site.

#### 2.2.2. Statistics

##### Statistical Analysis of the Primary Endpoint

In order to analyze and determine that at least one biomarker is above the limit of quantification, we will apply a binomial test for each biomarker to test for the proportion of patients with a value above the limit of quantification. Since a test against a null hypothesis of a proportion of exactly zero is not possible, the null value will be set to 0.001. The two-sided significance level will be set to 0.05. The result of the test will be interpreted in an exploratory manner. Further, we will calculate 95% confidence intervals for the proportion of patients above the limit of quantification.

##### Statistical Analysis of the Secondary Endpoint

(a)For all biomarkers, a Pearson correlation coefficient including a two-sided 95% confidence interval will be calculated. In the case of non-normally distributed values, the Spearman correlation coefficient will be used. If sufficient (i.e., ≥10), patients with dermal interstitial fluid levels above the limit of quantification will be available, and a linear regression with blood levels as the dependent and dermal interstitial fluid levels as the independent variable will be performed.(b)The proportion of patients with increased (difference to first/previous measurement ≥ 0) and decreased (difference to first/previous measurement < 0) values (blood level and dermal interstitial fluid levels) will be analyzed descriptively.(c)The difference in the proportion of patients with increased or decreased values (blood level and dermal interstitial fluid levels), either compared to the first or previous measurement, will be analyzed using a Boschloo test. The absolute numerical differences will also be analyzed using linear regression if the values for all patients are available. The difference in blood levels will be the dependent variable and the difference in dermal interstitial fluid levels as well as the blood levels from the first or previous measurement will be the two independent variables. Further, Pearson or Spearman correlation coefficients including 95% confidence intervals will be calculated for the differences in dermal interstitial fluid and blood levels.(d)A dISF sampling volume of at least 3 μL in at least 90% of the sampling procedures is defined as a fulfillment of the performance of the PELSA system according to its intended use. No dedicated statistical analysis is needed for an analysis of this endpoint.(e)The safety of the PELSA system in relation to its intended use is defined as the absence of the documentation of any device-related adverse event or serious adverse event. No dedicated statistical analysis is needed for the evaluation of this endpoint.

#### 2.2.3. Biomaterial Processing and Laboratory Analysis

##### dISF Processing

At each sampling time, dISF samples from both sampling sites are put together and aliquoted depending on the amount of dISF; see Figure A2 in Appendix B. Immediately after aliquoting, the samples are stored on dry ice and transferred to a −80 °C freezer as fast as possible.

##### Blood Processing

Heparin plasma is stored on ice after sampling until processing by a 15 min centrifugation at 2000× *g*, followed by pipetting and aliquoting. The time between sampling and centrifugation is limited to 120 min at a maximum. The aliquots are immediately put on dry ice until −80 °C storage.

EDTA blood is analyzed promptly without further processing.

##### dISF and Blood Analysis with Standard Laboratory Methods

Heparin plasma and dISF analysis are performed using Roche cobas^®^ assays (CRPHS, LACT2, and Elecsys proBNP II for hsCRP, lactate, and NT-proBNP, respectively). For technical reasons, the volume of the samples to be analyzed has to be at least 115 μL. Therefore, the dISF samples have to be diluted. The minimum volume of dISF for analysis is 10 μL, resulting in a 1:11.5 dilution. Sample volumes < 10 μL are only used for Olink^®^ analysis (at T0 and T2), where 1 μL is sufficient. The remaining dISF volume is used for the evaluation of a new biomarker sensor under development.

The blood count from EDTA blood is performed immediately using a Sysmex XN-9000 series (Sysmex, Kobe, Hyogo, Japan).

##### Olink^®^-Based dISF and Blood Analysis

Analysis using the Olink Target 96 Cardiovascular II panel (Olink Proteomics, Uppsala, Sweden) is performed on the dISF and plasma samples at T0 and T2. For the dISF analysis, a 1 μL dISF aliquot is diluted with 2 μL of Olink Sample Diluent. For blood analysis, processed heparin plasma aliquots are used. For a list of biomarkers analyzed by the aforementioned panel in Appendix C.

#### 2.2.4. Cardiopulmonary Exercise Testing

Bicycle ergometer CPET is performed using an Amedtec ECGpro ergometer (Amedtec, Aue, Germany) and Blue Cherry software (software version V1.3.0.501, Geratherm Respiratory, Bad Kissingen, Germany) according to the modified DZHK-SOP-K-07 [29], and using a ramp protocol. The start work load; steepness of the ramp; maximum work load; heart rate and blood pressure, both at rest and at maximum work load; respiratory exchange rate (RER); peak oxygen uptake (VO2); minute ventilation/carbon dioxide production (VE/VCO2) slope; and cause of limitation are documented.

#### 2.2.5. Measures for Quality Assurance

To assure a constant quality, the individual steps of biomaterial sampling, processing, and analysis, as described in this chapter, are harmonized and fixed in study-specific standard operating procedure (SOP) documents. Crucial details such as the timestamps of resting periods, biomaterial sampling, and freezing will be documented in a dedicated worksheet, which serves as source data. Data are documented in an electronic case report form (eCRF) using secuTrial^®^ software (software version V6.5.1.11, interActive Systems GmbH, Berlin, Germany). All analyzing laboratories are accredited by the competent authorities.

## 3. Results

### 3.1. Silicon Microneedle Technology as the Base of an Innovative dISF Extraction Device

Ascilion’s microneedles are manufactured in monocrystalline silicon in a 200 mm wafer (8″) MEMS fab. The body of the microneedles are defined by deep reactive ion etching and the sloping bevels by wet etching, resulting in ultra-sharp microneedle tips and bevel edges perfect for cutting the skin. The bore holes on the sloping bevel create a clog-free structure like hypodermic needles. There are 130 microneedles which are uniform on every chip (10 mm × 10 mm), with an average height of 450 ± 20 μm (Figure 8).

### 3.2. System/Device for dISF Extraction

The chip is mounted in a customized system, using negative pressure for improved penetration and increased sampling rates (Figure 9 and Figure 10).

The control and monitoring of the process is conducted through a graphical user interface on a computer, where the time and sub-pressure from the control unit and a video stream from the USB camera are presented. The real time image from the camera enables the operator to act if one or more pores from the microneedles produce blood, as well as to change the parameters if little or no dISF is produced in a certain time (Figure 11).

### 3.3. Protocol for dISF Extraction

Both skin penetration and extraction depend on well-controlled sub-pressure. The sub-pressure levels described below are the result of extensive pre-clinical tests, aiming at a balance between the highest achievable dISF amounts—usually achieved with a high sub-pressure level—and the lowest risk of blood contamination, skin irritation, and patient discomfort, which usually will be reached with lower sub-pressure levels. To minimize the risk for air leakage between the skin and components, the skin needs to be free from hair and the shaving of the sampling site is recommended. The sampling site is on the brachioradialis muscle, 80% of the distance from the radial styloid process and the antecubital fossa. During sampling, the microneedles are inserted ~0.3 mm into the skin for a few seconds by the application of a mild vacuum (−27 kPa relative atmospheric pressure). The Needle Unit is then removed from the tool and a Skin Fixture is placed around the sampling area and fastened to the arm using a wrist band. The Chamber Unit clicks into place on the Skin Fixture and a second vacuum protocol (−45 kPa) is initiated to bring dISF into the Chamber Unit. After 15 min, the camera is removed, the window of the Chamber Unit is removed, dISF is collected through the remaining aperture using a micropipette, and is dispensed into a 0.5 mL Eppendorf tube. The configuration and main phases of the protocol are depicted in the Figures below (Figure 12 and Figure 13).

The sampled volumes are in the microliter range and a high magnification camera is used for real-time visualization of the sampled fluid to support assessment of the volume and quality of the sample through the detection of blood contamination. The amount of blood contamination is assessed purely qualitatively. If blood is detected, the sampling is paused, blood is wiped away, and sampling is restarted after 2 min of waiting. If blood still emerges from the skin, a new attempt on a close-by area at −24 kPa is made using a new Needle Unit. If, on the other hand, no dISF is visible after three minutes, the penetration pressure is increased to −30 kPa. A detailed flow chart of the sampling is presented in Figure A1 in Appendix A.

A specific challenge, apart from developing a tool and method tolerable for the patient being sampled, and a limiting factor for dISF sampling to be competitive with blood sampling, is the amount of fluid that can be collected and used for further analysis. Hence, a semi-parallel procedure is developed where both arms of the subject could be used, and customized armbands (Skin Fixtures) are used to hold the devices in place (Figure 14).

### 3.4. A Comprehensive Study Design to Investigate Biomarkers in dISF

We present here a compact study designed to answer several questions around the use of dISF-derived biomarkers in heart failure patients. Heart failure is a leading cause of morbidity and mortality worldwide. One main treatment objective is to avoid the worsening of the disease state, leading to the affection of end-organ function and hospitalization in many cases, which worsens prognosis and causes high healthcare costs. Several attempts to intensify the monitoring of these patients, especially in the ambulatory setting, have shown improved outcomes. There are invasive methods, like continuous pulmonary artery pressure monitoring; multi-parametric algorithms derived from measurements (e.g., bioimpedance, heart rate, rhythm events) taken by cardiac implanted electronic devices, such as implantable cardioverter–defibrillators (ICDs) [30]; as well as telemonitoring using parameters like blood pressure, heart rate, and body weight monitoring [31]. Biomarker monitoring would have additional uses in these patients, but there is still no minimally invasive method to allow biomaterial sampling and analysis by the patients itself. Minimally invasive dISF sampling for biomarker analysis could close this gap, but there are several prerequisites for this. First, there has to be a proof that biomarkers can be detected in dISF; second, there has to be knowledge of the concentrations to be expected, and ideally a comparison conducted with the blood concentrations; third, the kinetics of dISF biomarker levels and their comparison with blood levels are of high interest, since these kinetics could be another sensitive marker of changing disease state; and fourth, sensor technology is needed to measure biomarkers from the very small dISF amounts that can usually be sampled.

This study is designed to answer these questions. Biomarkers from dISF and blood are measured using modified standard methods, i.e., standard analysis equipment with a protocol that includes dISF dilution to reach the biomaterial amounts necessary to run the laboratory machines. We focus on NT-proBNP as a well-established biomarker for heart failure diagnosis and prognosis estimation, lactate as a sensitive but less specific biomarker of changed circulatory conditions, and additionally hsCRP, taking into account that heart failure is an inflammatory condition too. Referring to already published data on the dISF levels of, e.g., electrolytes, lactate, and lipids in healthy subjects [1], we hypothesize that the dISF levels of the aforementioned biomarkers in heart failure patients are approximately in the same range as blood levels. The 1:11.5 dilution necessary for analysis with standard laboratory machines may affect detectability, especially of NT-proBNP, with its quite low plasma levels of about 10^2^ to 10^4^ pg/mL. On the other hand, in heart failure patients, NT-proBNP values in the upper measuring range can be expected, so we assume that detection will be achievable. For CRP, a high-sensitivity assay has been chosen to improve the chance of detection. To widen the number of analyzed biomarkers, we also use the Olink^®^ method, which allows analyses of a whole biomarker panel from as little as 1 μL of dISF. We use cardiocirculatory exercise test as a model, since at least lactate as well as NT-proBNP [32] kinetics can be expected during exercise. In the case of a sufficient amount of dISF, we will provide dISF aliquots to a partner within the DIGIPREDICT project to validate a new lactate sensor which works with amounts of as little as some μL of the dISF developed there.

## 4. Discussion

dISF is probably the most interesting alternative biofluid compared to blood for biomarker analysis, but overall, there is only limited data on dISF biomarkers so far. Furthermore, the extraction of sufficient amounts of dISF remains a main challenge that has impeded the broad use of dISF in biomedical research or even in clinical biomarker monitoring.

We describe here a study that has two important impacts: First, an innovative silicon microneedle chip-based device is presented that enables the sampling of pure dISF in a non-stigmatic, user-friendly manner with high reliability. The hollow microneedles enable well-controlled sub-pressure-assisted skin penetration. The sub-pressure is also used during the sampling and extraction of the dISF from the skin. The gradual ramping of sampling pressure allows the pores to open slowly and minimizes skin damage, as skin elasticity prevents rupture. The system’s performance is, on average, 1 microliter/minute during a 15 min period. With the parallel sampling method, 30 μL/patient and a sampling occasion is expected during the study.

Second, we introduce a comprehensive study design to investigate the detectability of cardiocirculatory and inflammatory biomarkers in dISF, examining their concentrations and investigating their kinetics during exercise in dISF and comparing this to those in blood. Furthermore, this study provides dISF for the validation of a new sensor adapted to lactate measurement from very small dISF amounts.

This study has several limitations. First, the sample size is small due to the exploratory design of the study, which may affect statistical power and lead to missing rare adverse events or device deficiencies. Second, there is no control group consisting of healthy subjects or other patient groups. Therefore, the study results will not be generalizable. Third, there is no structured assessment of skin properties, so it will not be possible to find the cause if there are large deviations in the acquired dISF amounts in single patients.

In conclusion, this study will foster research on biomarkers in dISF in general and in heart failure patients in particular, and thus will be an important step towards closer monitoring and personalized medicine, especially in high-risk patient groups. Further studies with larger participant groups and healthy controls are necessary to overcome the aforementioned limitations and gain more detailed knowledge in this field of research.

## Figures and Tables

**Figure 1 diagnostics-15-00989-f001:**
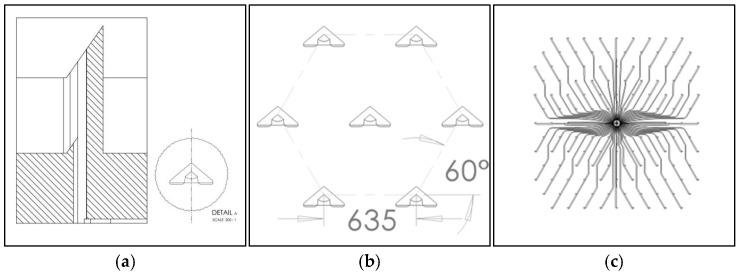
Design of ultra-sharp hollow microneedles in monocrystalline silicon. (**a**) Detailed cross section of a microneedle, its sloped bevel, and its slit and bore hole. (**b**) View of a unit cell on the microneedle chip. Each unit cell contains seven hollow needles. Each chip effectively contains 19 unit cells. (**c**) All needles are connected to each other by a capillary system on the back side of the chip (10 mm × 10 mm). Running dimensions in micrometers.

**Figure 2 diagnostics-15-00989-f002:**
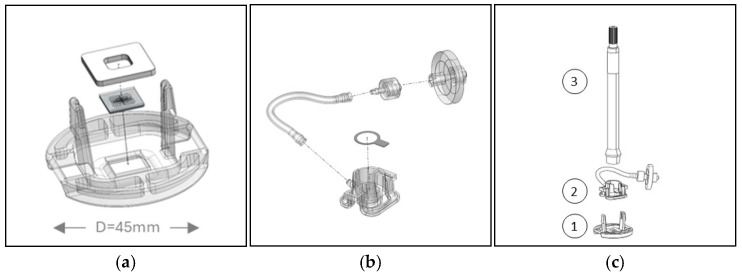
Design of the parts constituting the (**a**) Needle Unit and (**b**) Chamber Unit, and (**c**) the handheld part with the 1. Needle Unit, 2. Chamber Unit, and 3. Camera.

**Figure 3 diagnostics-15-00989-f003:**
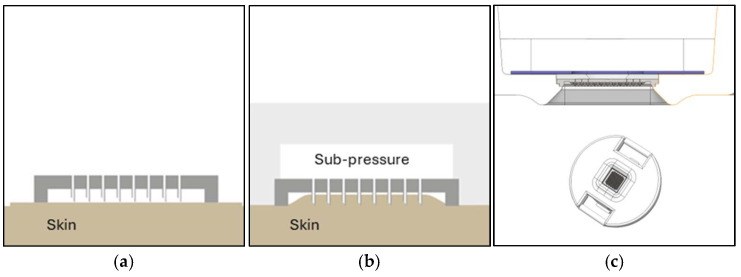
Schematic illustration of the sub-pressure-assisted penetration method. (**a**) Chip with hollow microneedles on the skin; (**b**) sub-pressure applied through hollow microneedles lifting the skin towards the base of the chip creating skin penetration; (**c**) cross section details of the Needle Unit. Please note the protrusion that forces the skin to stretch during sub-pressure.

**Figure 4 diagnostics-15-00989-f004:**
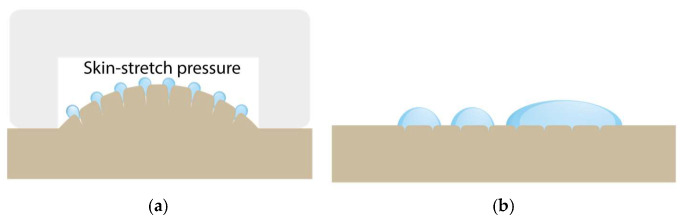
Schematic illustration of (**a**) sub-pressure-assisted skin stretching; (**b**) dISF droplets on the skin.

**Figure 5 diagnostics-15-00989-f005:**
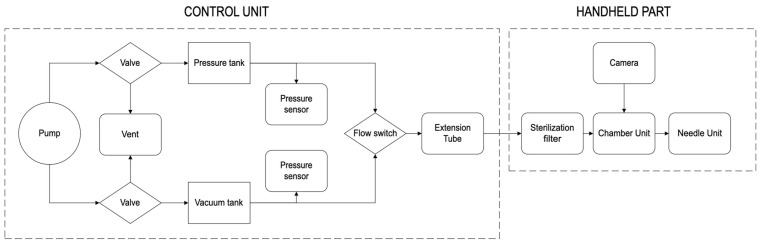
Schematic of the system design for the sampling of dermal interstitial fluid.

**Figure 6 diagnostics-15-00989-f006:**
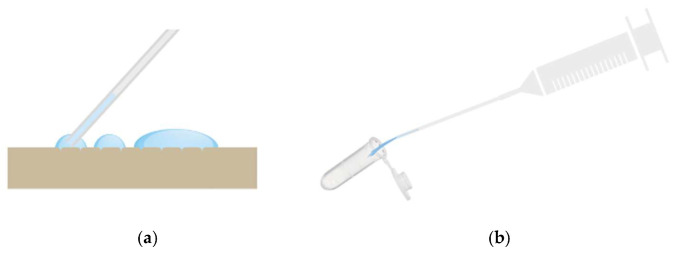
Schematics of the collection of dermal interstitial fluid. (**a**) Collection from skin and (**b**) dispensing into a storage container, e.g., an Eppendorf tube.

**Figure 7 diagnostics-15-00989-f007:**
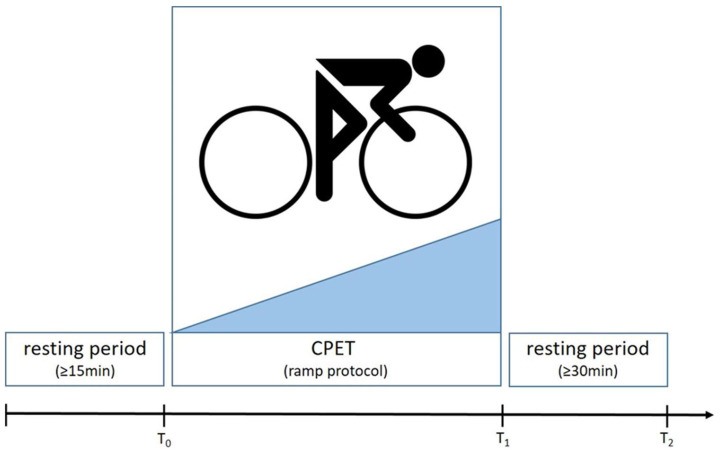
Schedule of CPET and biomaterial sampling.

**Figure 8 diagnostics-15-00989-f008:**
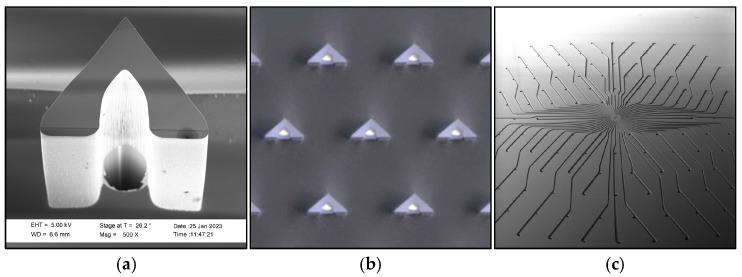
Photos of manufactured sharp hollow microneedles in monocrystalline silicon. (**a**) Scanning electron microscope picture of an individual microneedle. (**b**) Photo of a unit cell on the microneedle chip. Each unit cell contains seven hollow needles. Each chip effectively contains 19 unit cells. (**c**) All needles are connected to each other by a capillary system on the back side of the chip. The capillaries are created by deep reactive etching of the silicon chip. Compare this to Figure 1.

**Figure 9 diagnostics-15-00989-f009:**
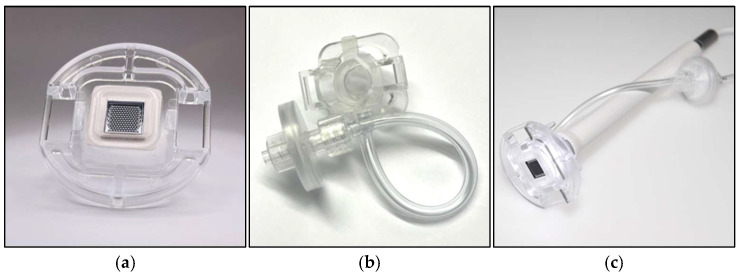
Manufactured parts: (**a**) Needle Unit, (**b**) Chamber Unit, and (**c**) handheld part, including the camera for optical inspection. Compare this to Figure 2.

**Figure 10 diagnostics-15-00989-f010:**
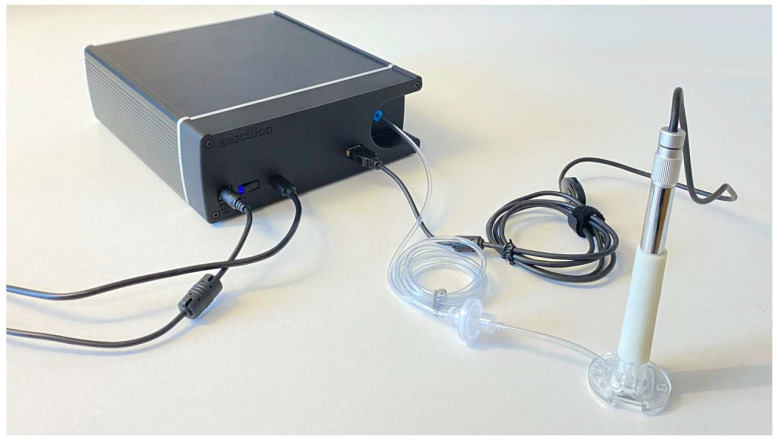
Photo of manufactured system to be used on HF patients. Compare this to Figure 5.

**Figure 11 diagnostics-15-00989-f011:**
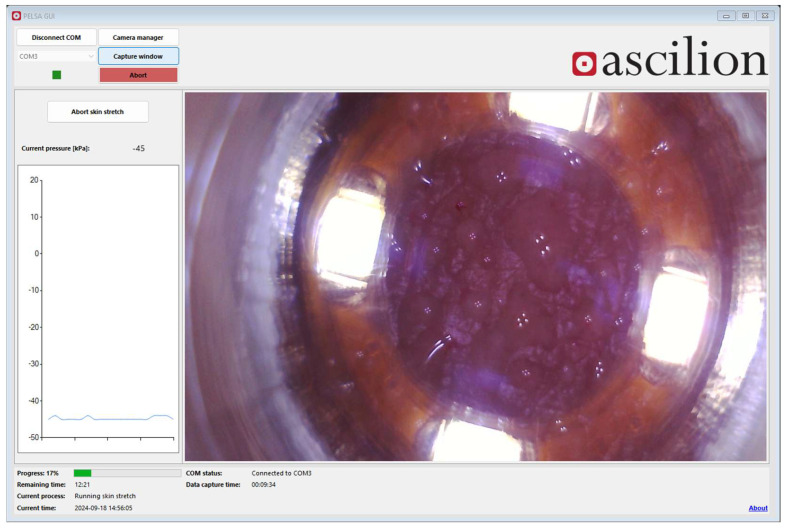
Picture of the graphical user interface controlling the vacuum and monitoring the sampling process. Droplets of dISF are visible on screen through the camera system.

**Figure 12 diagnostics-15-00989-f012:**
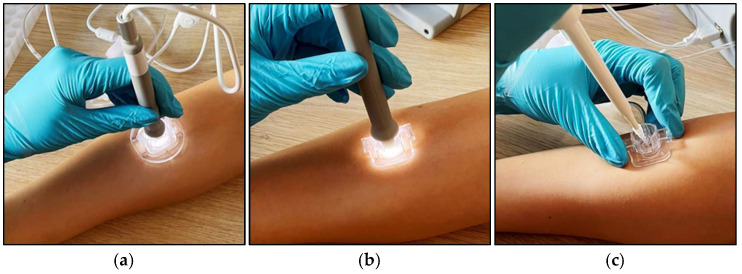
Photos of the system configuration during the three main phases: (**a**) skin penetration, (**b**) extraction, and (**c**) collection.

**Figure 13 diagnostics-15-00989-f013:**
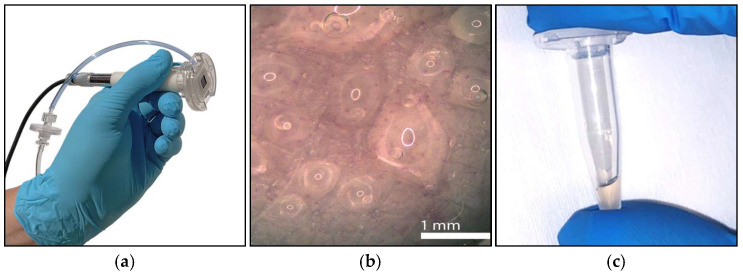
(**a**) The handheld part of the dISF sampling device, including the sub-pressure tubing, sterile filter, and camera. (**b**) View of dISF on the skin through the on-board camera system in the handheld. (**c**) Transparent dermal interstitial fluid (38 microliter) sampled and collected during 15 min of training at DHZC and dispensed into an Eppendorf tube.

**Figure 14 diagnostics-15-00989-f014:**
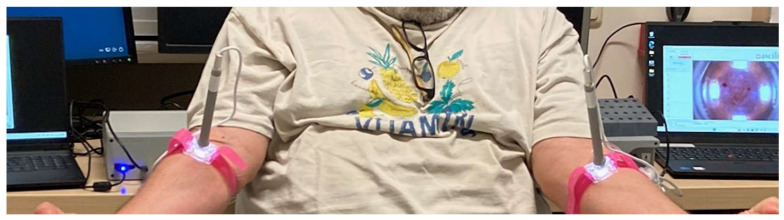
Photo of dual sampling on a healthy volunteer during dress rehearsal where 2 × 15 microliters of dISF was collected.

**Table 1 diagnostics-15-00989-t001:** Inclusion and exclusion criteria.

Inclusion Criteria	Exclusion Criteria
- Subject suffers from heart failure (according to ESC 2021 Heart Failure Guidelines). - Age of subject is ≥18 years. - Subject is female or male. - For female subjects: Confirmed post-menopausal state, defined as amenorrhea for at least 12 months; If having childbearing potential: Negative highly sensitive urine or serum pregnancy test before inclusion; Practicing a highly effective birth control method (failure rate of less than 1%): combined (estrogen- and progestogen-containing) hormonal contraception associated with inhibition of ovulation (oral/intravaginal/transdermal); progestogen-only hormonal contraception associated with inhibition of ovulation (oral/injectable/implantable); intrauterine device (IUD); intrauterine hormone-releasing system (IUS); bilateral tubal occlusion;vasectomized partner; heterosexual abstinence. - Subject is capable of performing cardiopulmonary exercise testing.	- Subject is breastfeeding. - Subject suffers from an addiction or from a disease that prevents the subject from recognizing the nature, scope, and consequences of the study. - Subject is treated with immunosuppressive drugs at enrolment. - Subject requires renal replacement therapy. - Subject has a known colonization or infection with multi-drug-resistant pathogens. - Subject has an open wound in or near the sampling area. - Subject has any type of tattoo or piercing anywhere in or near the sampling area. - Subject shows an inability to comply with all the study procedures and follow-up visits. - Subject is unwilling to consent to the saving and propagation of pseudonymized medical data for study reasons. - Subject is legally detained in an official institution. - Subject is dependent on the sponsor, the investigator, or the study site. - Subject participates in a study according to AMG/CTR that investigates immunosuppressive or anticoagulant drugs at the time of this study.

## Data Availability

The original contributions presented in this study are included in the article. Further inquiries can be directed to the corresponding author(s).

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
