# Peer review of "Use of a Silicon Microneedle Chip-Based Device for the Extraction and Subsequent Analysis of Dermal Interstitial Fluid in Heart Failure Patients"

_diagnostics, 2025, doi:10.3390/diagnostics15080989_

Round 1
Reviewer 1 Report
Comments and Suggestions for Authors
Dear Authors,
I congratulate you on your paper. Your research and the development of a new technique for microneedling to recover dermal interstitial fluid are very innovative, and the device presented could be of great use in medical practice.
The introduction and materials and methods sections of the paper are very well written; however, the results and conclusions are concerning.
The article does not clearly present the results comparing the values of biomarkers in both blood and dermal fluid. Furthermore, it does not provide clear data regarding the efficacy of recovering dermal fluid, its quality, and the ability to determine biomarkers and other parameters from it.
Please review these issues before resubmitting the paper.
Kind regards
Author Response
Dear Reviewer,
We want to thank you a lot for the review of our work and for your valuable comments.
Regarding your totally correct remark that we do not clearly present the results of biomarker levels in blood and in dISF as well as data regarding the efficacy of recovering dermal interstitial fluid with the silicon microneedle-based PELSA system introduced here, we have to admit that this paper focusses on the dISF extraction method and device, and on the description of the protocol for a proposed study, but does not yet intend to present the results of the study still to be conducted.
Yours sincerely,
Laurenz Kopp Fernandes and Markus Renlund
Reviewer 2 Report
Comments and Suggestions for Authors
Dear Sirs,
I would like to thank you very much for sending me the study entitled:
„Use of a silicon microneedle chip-based device for extraction and subsequent analysis of dermal interstitial fluid in heart failure patients.”
I warmly congratulate the authors of the manuscript.
Heart failure is one of the predominant problems in modern cardiology. Early detection of cardiovascular destabilisation allows appropriate therapy to be implemented before full circulatory decompensation occurs. The method presented may be a useful tool in the coordinated therapy of patients with heart failure. It also seems to me that the material presented will find interest among the readers of the journal.
Yours sincerely,
Reviewer
Author Response
Dear Reviewer,
We want to thank you a lot for the review of our work and for your comments.
Yours sincerely,
Laurenz Kopp Fernandes and Markus Renlund
Reviewer 3 Report
Comments and Suggestions for Authors
An interesting research program that can be helpful in the diagnosis of heart failure, detection of the stage of the disease, and therefore treatment procedures. However, this is a preliminary report and is characterized by a small study group, so in order to draw further conclusions, it is necessary to continue the research, which is also emphasized by the authors.
I believe that the article will bring interesting information and should be accepted without correction.
Author Response

(The authors gave the same response as above.)

Reviewer 4 Report
Comments and Suggestions for Authors
The authors investigated the use of silicon microneedles to extract dISF and assess its potential for biomarker analysis in heart failure patients. Below are my concerns and suggestions:
- Comprehensive comparison with conventional techniques.
- Clearly summarize biomarker concentrations (e.g., expected vs. observed in dISF).
- Healthy controls? Has reproducibility been tested? Potential skin variability?
- Quantify blood contamination.
- The relationship between sub-pressure levels and sample volume?
- The timing of dISF sampling was within 12 minutes post-exercise, but was inconsistent, this variability may introduce bias due to the rapid changes in biomarker kinetics.
- Any controls for exercise-induced changes in hemoconcentration-related biomarkers?
- Whether sample dilution affects biomarker detectability?
- The sample size is small.
Author Response
Dear Reviewer,
We want to thank you a lot for the review of our work and for your very valuable comments that we all have addressed in the revised version.
- To point out the comparison with conventional, established methods for dISF recovery, we have adapted the regarding part of the introduction (page 2, lines 51-57)
- and 8. We have addressed the very important issues of expected biomarker concentrations, of sample dilution, and of its possible influence on detectability in 3.4 (pages 14/15, lines 414-422) now. To shortly summarize, based on already published data on dISF biomarker levels in other patient groups, we expect that it will be possible to detect the biomarkers examined here in our dedicated patient group and with the proposed methods. Regarding observed dISF concentrations, we have to admit that this paper focusses on the dISF extraction method and device and on the description of a proposed study protocol, but does not yet intend to present the results of the study still to be conducted.
- and 9. You are totally correct that the lack of healthy controls and the small sample size are relevant limitations of the planned study as well as skin variability may influence dISF extraction. We have clearly named these limitations in the discussion (page 15, lines 449-454) now.
- We fully agree that quantification of blood contamination is an important issue. We have added a clarification on this issue in 3.3 (page 13, lines 370/371), where also is described that blood contamination will cause pausing of the sampling and adaption of the sampling protocol. The flowchart in Annex A (Fig. A.1) visualizes this.
- We confirm that there is a meaningful relationship between sub-pressure levels and sample volume which has been subject to extensive pre-clinical tests. We have added a clarifying section in 3.3 (pages 12/13, lines 344-348)
- This is a very important remark that we have addressed now in 2.2.1 (page 7, lines 207-212), explaining the choice of the sampling time as well as measures taken to assure quality of the results.
- The influence of hemoconcentration is indeed an important issue when measuring biomarkers during exercise and we thank you for pointing this out here. We address this problem normalizing biomarker levels for plasma volume as described in 2.2.1 (page 7, lines 212-216).
Yours sincerely,
Laurenz Kopp Fernandes and Markus Renlund
Reviewer 5 Report
Comments and Suggestions for Authors
The article and the proposed study explains very clearly the usselfulnes and the challenges of using Silicon Microneedle Chip-based Device for Extraction and Subsequent Analysis of Dermal Interstitial Fluid in Heart Failure Patients. However it seems difficult to extract enough volume due to the filtration effect that lowers concentration of the analytes in the collected samples. No data coming for healthy volunteers are mentioned in order to predict the correlation of proposed biomarkers beetwen DIF and blood size. Therefore, please explain this two aspects in order to better understand the reliability of the investigation. Finally, I wish to congratulate the authors for their impressive research that could enhance the diagnostic and folow-up of patients with heart failure
Author Response
Dear Reviewer,
We want to thank you a lot for the review of our work and for your very valuable comments.
You are fully correct in that dermal interstitial fluid is the product of blood plasma that passes through the vessel wall and thereby is filtered. Here we plan to investigate the correlation between biomarker concentration in plasma and dermal interstitial fluid, which is expected to be influenced by several factors, of which true physiological concentrations being perhaps the most important one. However, it has previously been shown that many biomarkers have concentrations that correlate well between blood plasma and dermal interstitial fluid [References 2, 3] and in some cases with increased prevalence in ISF [Reference 3]. Therefore, it seems probable that some markers will be found in high enough concentrations in ISF to lead to important progress within the field. To point this out better, we have completed the introduction (page 2, lines 48-50) as well as chapter 3.4 (pages 14/15, lines 414-422).
Yours sincerely,
Laurenz Kopp Fernandes and Markus Renlund
Round 2
Reviewer 4 Report
Comments and Suggestions for Authors
The authors have addressed some of the previous comments, however, simply discussing the concerns or limitations is not sufficient, the data provided and the resulting conclusions remain insufficiently supported.
Author Response
Dear editor, dear reviewers,
Thank you for all the review feedback. We realize that we could clarify a few things in order to address the concerns mentioned.
First of all, we should clarify that we focus here on the dISF extraction method and device and on the description of a proposed study protocol, but do not yet intend to present the results of the study still to be conducted. The study aims to detect cardiovascular biomarkers in the dISF samples to be taken, which is an important step towards future diagnostic methods. It will be a very early stage clinical study aiming at finding cardiovascular biomarkers in a relatively unexplored sample type (dISF). To address the concerns on this, we have completed the introduction be another explaining last sentence (see lines 61-64).
As mentioned in the article, many attempts to sample dISF have been done, but it is extremely difficult to sample this kind of fluid in a way and amount that it can be used in a clinical setting. The device described in the paper has led to faster dISF sampling leading to provide enough volume for certain analysis methods, and is one of the first devices really suitable for clinical studies with pure dISF.
Furthermore, current laboratory methods usually need much higher biomaterial volumes than the µL range that can usually be acquired by current dISF sampling methods. As described in detail in the article, we will use two different laboratory methods, (1) an adapted standard laboraty approach working with dISF dilution, and (2) the Olink method.
To our knowledge, detectability and concentrations of most of the biomarkers (the Olink cardiovascular panel II-assessed biomarkers included) that we focus on in our study have never been investigated in dISF before for the reasons mentioned above. It is of course important to understand that the concentrations of target biomolecules could be too low in dISF, especially when it is diluted, but exactly since information on this is limited, investigation of these biomarkers in dISF is one main objective of this planned proof-of-concept study that we describe here. Reproducibility on the other hand will only be able to be tested in a second study, when data of this first exploratory study will be available.
Another important issue is that several of the examined biomarkers only appear in very low concentrations in healthy subjects. Therefore we have chosen a patient cohort with manifest cardiac disease for this first exploratory study to increase the change of biomarker detectability. We totally agree that comparison with healthy subjects is important and should be subject to subsequent research.
We know that the proposed study will only be a first step in this field of investigation and has several limitations, so it will probably be necessary to conduct subsequent larger studies, as we have discussed in detail in Section 4.
We really hope that these statements are able to allay your concerns.
With best regards,
Laurenz Kopp Fernandes and Marcus Renlund
Round 3
Reviewer 4 Report
Comments and Suggestions for Authors
The authors clarified the aims of the study, which helps to minimize potential misinterpretation of the results.